# Corrupted adipose tissue endogenous myelopoiesis initiates diet-induced metabolic disease

Elodie Luche[1], Virginie Robert[1], Vincent Cuminetti[1], Celine Pomié[2], Quentin Sastourné-Arrey[1], Aurélie Waget[2], Emmanuelle Arnaud[1], Audrey Varin[3], Elodie Labit[1], Patrick Laharrague[1], Remy Burcelin[2†], Louis Casteilla[1†], Beatrice Cousin[1]*

[1]STROMALab, Université de Toulouse, CNRS ERL 5311, EFS, INP-ENVT, Inserm U1031, UPS, Toulouse, France; [2]INSERM U1048, Université de Toulouse, EFS, INP-ENVT, Inserm U1031, UPS, Toulouse, France; [3]STROMALab, Université de Toulouse, EFS, INP-ENVT, Inserm U1031, UPS, Toulouse, France

*For correspondence: beatrice.
cousin@inserm.fr

†These authors contributed
equally to this work

Competing interests: The
authors declare that no
competing interests exist.

Reviewing editor: Michael
Czech, University of
Massachusetts Medical School,
United States

**Abstract** Activation and increased numbers of inflammatory macrophages, in adipose tissue (AT) are deleterious in metabolic diseases. Up to now, AT macrophages (ATM) accumulation was considered to be due to blood infiltration or local proliferation, although the presence of resident hematopoietic stem/progenitor cells (Lin-/Sca+/c-Kit+; LSK phenotype) in the AT (AT-LSK) has been reported. By using transplantation of sorted AT-LSK and gain and loss of function studies we show that some of the inflammatory ATM inducing metabolic disease, originate from resident AT-LSK. Transplantation of AT-LSK sorted from high fat diet-fed (HFD) mice is sufficient to induce ATM accumulation, and to transfer metabolic disease in control mice. Conversely, the transplantation of control AT-LSK improves both AT-inflammation and glucose homeostasis in HFD mice. Our results clearly demonstrate that resident AT-LSK are one of the key point of metabolic disease, and could thus constitute a new promising therapeutic target to fight against metabolic disease.

## Introduction

Over the last two decades a critical role of adipose tissue macrophages (ATM) in the initiation of the metabolic inflammation, commonly found in highly prevalent chronic diseases such as obesity, has been largely documented (*Chawla et al., 2011*). At the onset of high-fat diet (HFD)-induced metabolic disease, accumulating ATM exhibit an inflammatory phenotype characterized by CD11c expression and this contributes to adipocyte dysfunction and insulin resistance (*Lumeng et al., 2007*; *Lumeng et al., 2008*). In addition, an increased number of ATM worsens features of metabolic disease (*Hirasaka et al., 2007*; *Kamei et al., 2006*), whereas a reduced number of ATM is associated with a reduced metabolic inflammation and an improved metabolic status (*Kanda et al., 2006*). Although most of the literature supports the idea that ATM cause disease, accumulating evidences recently show that these ATM also have beneficial effects in obesity. Different molecular tools such as lipid storage (*Aouadi et al., 2014*), autophagy (*Fitzgibbons and Czech, 2016*), or program of lysosomal metabolism (*Xu et al., 2013*) are used by the ATM to buffer and process excessive amounts of lipids and maintain adipose tissue (AT) homeostasis, in the context of metabolic disease. In addition, an acute inflammatory response in the AT would be essential for adipose remodeling and adipocyte differentiation, and thus for a healthy AT expansion (*Cinti et al., 2005*; *Wernstedt Asterholm et al., 2014*; *Ye and McGuinness, 2013*).

It is of common knowledge that macrophages invading the AT originate from the bone marrow (BM) via blood circulation (*Weisberg et al., 2003*) notably in response to AT inflammation which has been shown to trigger the proliferation of BM hematopoietic progenitors and myelopoiesis, that in turn perpetuate local maladaptive inflammatory response (*Nagareddy et al., 2014, 2013*; *Singer et al., 2014, 2015*). In addition to monocyte recruitment, recent studies suggest a local regulation and proliferation of ATM, independently of BM precursors. Reduced apoptosis, local proliferation, as well as reduced migration capacity of resident ATM may also lead to their accumulation in obesity (*Amano et al., 2014*; *Hill et al., 2015*; *Ramkhelawon et al., 2014*).

In physiological state, we and others recently demonstrated the presence of a peculiar functional resident hematopoietic stem/progenitor cells (LSK) population in the AT (*Han et al., 2010*; *Poglio et al., 2012, 2010*) that can renew innate immune cells and especially macrophages in the AT, via in situ differentiation. We thus hypothesized that accumulating ATM in metabolic disease could partly originate from AT-endogenous hematopoietic activity. To study the role of AT-LSK in the initiation of metabolic inflammation, we transplanted the corresponding cells from diabetic or control mice to their counterpart recipient mice, exploring therefore their capacity to transfer the metabolic phenotype and demonstrate their causal role. We show here that some of the inflammatory ATM that induce metabolic disease development, originate from resident AT-LSK, demonstrating that AT-LSK are one of the key point of metabolic disease. Altogether our results provide a new mechanism for the increase in pro-inflammatory ATM in metabolic diseases and point out AT-LSK as novel regulator of these pathologies.

## Results

### Inflammatory ATM derived from subcutaneous (sc) AT-LSK differentiation accumulate in HFD mice

To investigate whether some of the inflammatory macrophages in the AT could originate from AT-LSK during metabolic disease, chimeric mice were generated by using standard repopulation assays as previously described (*Poglio et al., 2012*) by injecting $2.10^3$ sorted scAT-LSK mixed with $2.10^5$ congenic BM cells isolated from congenic CD45 variants. Injection of BM cells ensure the survival of the recipient, as previously described (*Miller et al., 2008*). Two months after transplantation, chimeric mice were fed either a normal chow (NC) or a HFD during 12 weeks (*Figure 1a*). This diet has been previously described to induce insulin resistance and dysglycemia in mice (*Burcelin et al., 2002*; *Fernández-Real et al., 2011*).

We first characterized the metabolic and inflammatory profiles of these chimeric mice. As expected, after 12 weeks on HFD, body weight (*Figure 1b*) and scAT weight (*Figure 1c*) were slightly increased. Chimeric mice became glucose intolerant as revealed by intraperitoneal glucose tolerance test (ipGTT) (*Figure 1d*), their fasting glucose and insulin levels were significantly higher from that of normal chow fed mice (*Figure 1d and e*), and their scAT exhibited insulin resistance as shown by reduced glucose utilization in response to insulin (*Figure 1f*). In HFD chimeric mice, the ATM number (identified as F4/80$^+$/MHCII$^+$) significantly increased in the scAT (*Figure 1g*), and was associated with a higher expression of inflammatory cytokines (Il-1b, PAI1 and CCL2) (*Figure 1h*), confirming that HFD induces a metabolic inflammation in the scAT of chimeric mice.

To assess the reconstitution efficiency, the chimerism (percentage of CD45.1$^+$ donor cells among total CD45$^+$ cells) was determined by flow cytometry in the scAT (*Figure 1i*) and the BM (*Figure 1j*), as well as in other metabolic tissues such as visceral AT (pgAT), liver and muscles. The chimerism was similar in the scAT in both control and HFD chimeric mice, whereas in the pgAT it was slightly decreased in HFD group compared to NC (*Figure 1k*). In the liver and muscles, the chimerism was very low (*Figure 1k*), suggesting no significant impact of CD45.1$^+$ cells in these organs. As expected and previously described (*Poglio et al., 2012*), no CD45.1$^+$ cells were identified in the BM (*Figure 1k*). To validate however the animal model of hematopoietic reconstitution, chimerism was also quantified in chimeric mice obtained after transplantation of $2.10^3$ sorted BM-LSK mixed with $2.10^5$ congenic BM cells isolated from congenic CD45 variants. In these mice, CD45.1$^+$ cells were observed in both the scAT and the BM as expected (*Poglio et al., 2012*), and the chimerism was not significantly modified by the diet (*Figure 1—figure supplement 1*).

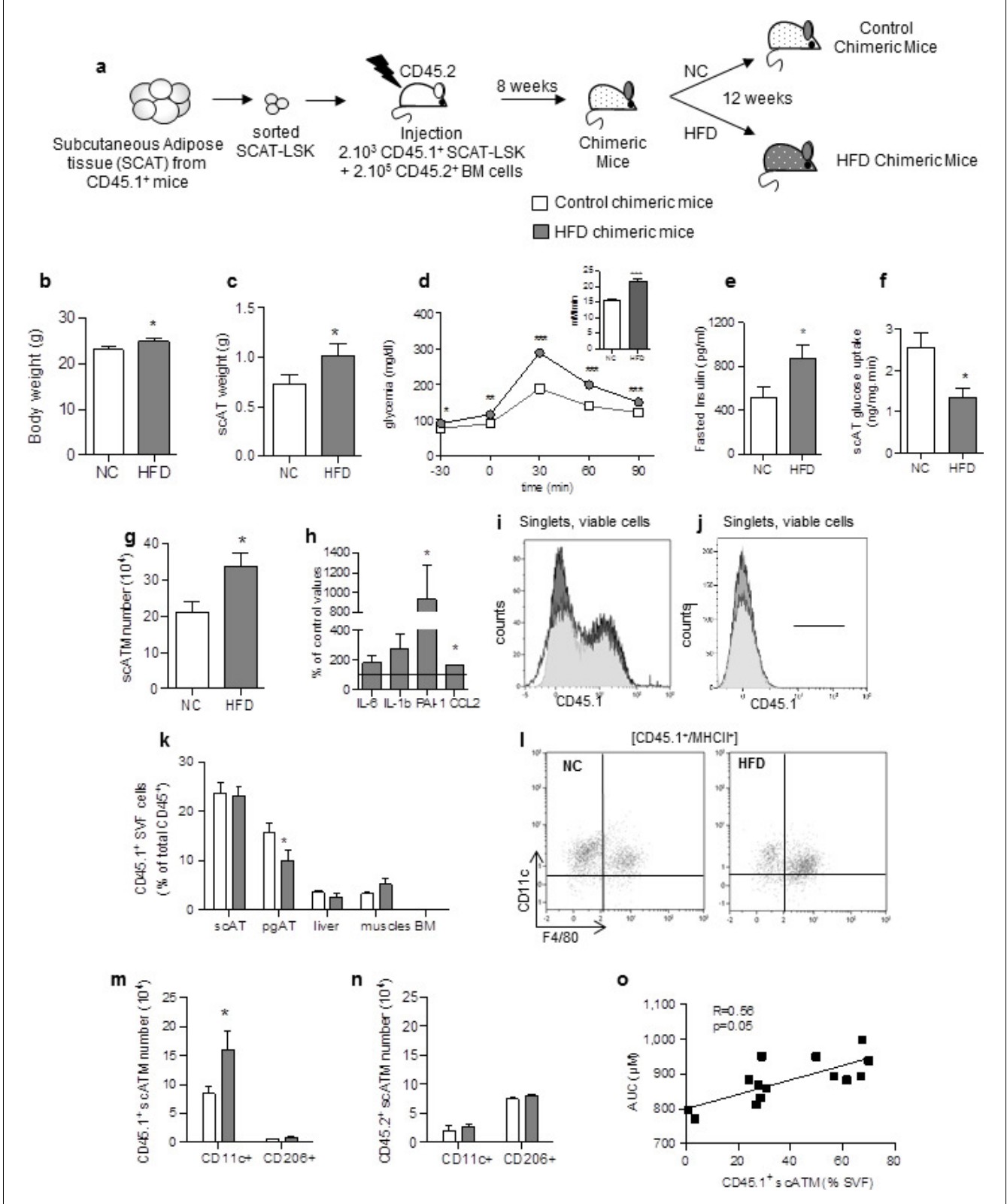

**Figure 1.** Accumulating subcutaneous Adipose Tissue Macrophages (scATM) in HFD-chimeric mice originate from scAT-LSK. (a) Schematic representation of experimental procedure: lethally irradiated CD45.2[+] recipients mice were co-injected with sorted scAT-LSK and total BM cells isolated respectively from CD45.1 and CD45.2 expressing donor mice, and were then fed a normal Chow (NC) or a high fat diet (HFD) for 12 weeks. Metabolic profile was investigated in chimeric mice by (b) body weight, (c) scAT weight, (d and insert) IPGTT and AUC, (e) quantification of fasted insulin and (f)

*Figure 1 continued on next page*

*Figure 1 continued*

scAT-glucose uptake (ng/mg.min). (**g**) Quantification of ATM identified as F4/80$^+$/MHCII$^+$ in the scAT. (**h**) Expression of genes encoding for inflammatory cytokines analyzed by qRT-PCR in the scAT and expressed as a percent of control values obtained in NC mice. (**i–n**) Flow cytometry was performed on SVF and BM cells from NC and HFD mice to identify CD45.1$^+$ populations. Representative histograms of CD45.1$^+$ cells in the scAT (**i**), and the BM (**j**). (**k**) Total chimerism in the scAT, pgAT, liver, muscles and BM expressed as percent of total CD45$^+$ cells. (**l**) Representative dot plots of flow cytometry analysis showing CD11c$^+$/F4/80$^+$/MHCII$^+$ cells in a CD45.1$^+$ cell population gated on singlet live cells. Quantification of scAT-LSK-derived (**m**) and BM-derived (**n**) pro-inflammatory (MHCII$^+$/F4/80$^+$/CD11c$^+$) and anti-inflammatory scATM populations (MHCII$^+$/F4/80$^+$/CD206$^+$) expressed in absolute numbers. (**o**) Correlation between CD45.1+ ATM content (expressed in % of SVF) and AUC. Results are expressed as mean ± sem of 4 to 26 individual animals in control (white symbols) and HFD (grey symbols) groups. Comparisons between groups were made with the nonparametric Mann-Whitney test. *p<0.05; **p<0.01; ***p<0.001.

The following figure supplements are available for figure 1:

**Figure supplement 1.** Validation of the animal model of hematopoietic reconstitution.

**Figure supplement 2.** Absence of accumulation of LSK-derived ATM in the pgAT of chimeric HFD mice.

The origin and the nature of ATM were then determined by analysis of F4/80, MHCII, CD11c and CD206 expression in CD45.1 population (*Figure 1l*). After 12 weeks of HFD, the population of ATM deriving from scAT-LSK (CD45.1$^+$) significantly increased in the scAT, accounting for 56 ± 6% of total scAT-ATM. This accumulation of ATM in the scAT was mainly and specifically due to an increase in inflammatory CD45.1$^+$/CD11c$^+$ macrophage number without any change in anti-inflammatory CD45.1$^+$/CD206$^+$ ATM content (*Figure 1m*). In addition, no change in pro- and anti-inflammatory ATM originating from the BM, was observed in the scAT of HFD fed chimeric mice (*Figure 1n*). In the pgAT, both CD11c$^+$ and CD206$^+$ ATM numbers originating from donor mice were similar in NC and HFD mice (*Figure 1—figure supplement 2a*). Accordingly, the expression of pro-inflammatory cytokines in pgAT was not significantly modified by the HFD (*Figure 1—figure supplement 2b*). Finally a significant positive correlation between scAT CD45.1$^+$ ATM content (derived from scAT-LSK) and glucose intolerance was drawn (*Figure 1o*) suggesting a relationship between scAT-LSK hematopoietic activity and metabolic disease.

These results showed that HFD-induced diabetes is associated with an over production of inflammatory ATM in the scAT, originating in part from scAT-LSK differentiation.

## ATM derived from scAT-LSK accumulate in AT of HFD mice due to early proliferation of progenitor cells

The increase in donor ATM in HFD chimeric mice could be due to increased proliferation and/or differentiation of progenitor cells, and/or to proliferation of mature macrophages. LSK and myeloid progenitors were therefore quantified in SVF cells from NC and HFD chimeric mice, based on the phenotype described previously (*van Galen et al., 2014*) and macrophage proliferation was assessed by staining with an antibody against the proliferation marker Ki67 (*Amano et al., 2014*). The numbers of CD45.1$^+$ LSK cells (*Figure 2—figure supplement 1a*), CD45.1$^+$Multipotent Progenitors (MPP) (*Figure 2—figure supplement 1b*), and CD45.1$^+$ Common Myeloid Progenitors (CMP) (*Figure 2—figure supplement 1c*), and the percentage of proliferating (Ki67$^+$) ATM (*Figure 2—figure supplement 1d*) were similar in chimeric mice fed a HFD for 3 months, compared to NC. These results suggest that the dynamic phase of the scAT ATM accumulation occurred earlier in the pathogenesis. CD45.1$^+$ LSK cells and myeloid progenitors were therefore quantified in chimeric mice fed a HFD for 3†7 days. As shown in *Figure 2*, a few days of HFD were sufficient to induce an increase in CD45.1$^+$ LSK (*Figure 2a*), MPP (*Figure 2b*) and CMP (*Figure 2c*) numbers, whereas the number CD45.1$^+$ scATM was not yet significantly modified (*Figure 2d*). This accumulation of CD45.1$^+$ LSK and myeloid progenitor cells in the scAT of HFD mice was due to an enhanced proliferation as revealed by Ki67 staining (*Figure 2e*).In contrast, the proliferation of CD45.1$^+$ ATM was low and unchanged by the diet (*Figure 2f*).

These results demonstrate that accumulation of ATM deriving from scAT-LSK in the scAT of HFD mice is due to an increase in myeloid progenitor proliferation that occurs very early in the diabetes development.

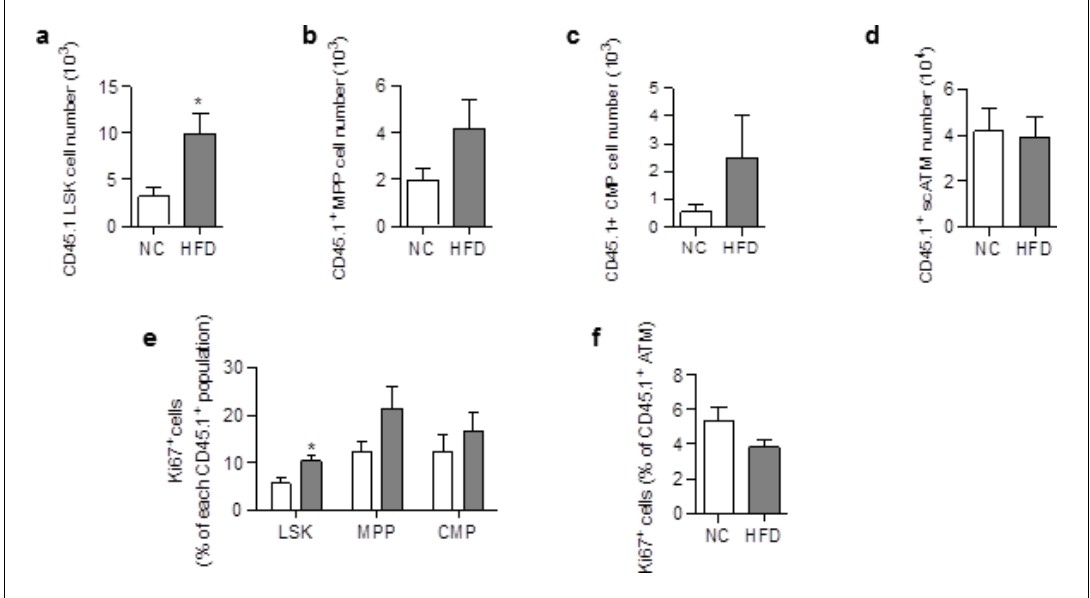

**Figure 2.** ATM accumulation in the scAT of HFD mice is due to enhanced proliferation of myeloid progenitors at the onset of the disease. Flow cytometry was performed on SVF cells from mice fed a NC or a HFD for 3 to 6 days, to identify myeloid progenitors and ATM. Quantification of LSK cells (a); Lin$^-$/Sca-1$^+$/c-Kit$^+$), Multipotent Progenitors (b); MPP; Lin$^-$/Sca-1$^+$/c-Kit$^+$/CD34$^+$); Common Myeloid Progenitors (c); CMP; Lin$^-$/Sca-1$^-$/c-Kit$^+$/ CD34$^+$) and ATM (d); MHCII+/F4/80+) within the CD45.1$^+$ population, expressed in absolute numbers. Proliferation was assessed by Ki67 staining in CD45.1$^+$ LSK and myeloid progenitors (e) as well as in CD45.1$^+$ ATM (f), and expressed as percent of each CD45.1$^+$ population. Results are expressed as mean ± sem of 4 to 8 individual animals in control (white symbols) and HFD (grey symbols) groups. Comparisons between groups were made with the nonparametric Mann-Whitney test. *p<0.05.

The following figure supplement is available for figure 2:

**Figure supplement 1.** Differentiation and proliferation of ATM are unchanged in the scAT of chimeric mice after 3 months of HFD.

## Specific depletion of ATM derived from scAT-LSK improves glucose metabolism

To confirm the role of inflammatory scATM derived from scAT-LSK in metabolic inflammation, an hematopoietic repopulation assay was performed, using scAT-LSK isolated from CD45.2$^+$ mice expressing the diphtheria toxin receptor (DTR) under control of the full CD11c promoter (*Patsouris et al., 2008*), and transplanted into lethally irradiated CD45.1$^+$ recipient mice. After 12 weeks of HFD, chimeric mice received diphtheria toxin (DT) in order to specifically remove CD11c$^+$ ATM deriving from scAT-LSK (*Figure 3a*).

Treatment with DT did not significantly modify body- and scAT weights (*Figure 3b and c*). Reconstitution efficiency was measured by flow cytometry in the scAT (*Figure 3d*). As expected, a significant decrease in CD45.2$^+$ cell number was observed after DT-treatment due to removal of CD45.2$^+$ ATM. Indeed, chimerism reached 67.8 ± 1% in the scAT of HFD mice versus 42.3 ± 3.5% of total CD45$^+$ cells in the scAT of HFD mice treated with DT (p=0.015, nonparametric Mann-Whitney test). No CD45.2$^+$ cells were detected in the BM (*Figure 3e*). DT treatment induced a specific and drastic depletion in inflammatory scATM derived from scAT-LSK (CD11c$^+$/F4/80$^+$/MHCII$^+$/CD45.2$^+$) (*Figure 3f and g*),but not in those derived from the BM (*Figure 3h*). This was associated with a significant decrease in the expression of inflammatory cytokines compared to HFD non treated mice (*Figure 3i*). CD11c is also highly expressed in dendritic cells. A slight decrease in DC number originating from scAT-LSK was observed after DT treatment in the scAT whereas BM-derived DC number remained unchanged (*Figure 3j and k*). Since HFD did not modify donor-derived DC number in the scAT (9.2 ± 2.6 10$^3$ in NC vs 11.5 ± 2.1 10$^3$ in HFD group), it is however unlikely that the decrease observed after DT treatment plays a key role in the pathogenesis.

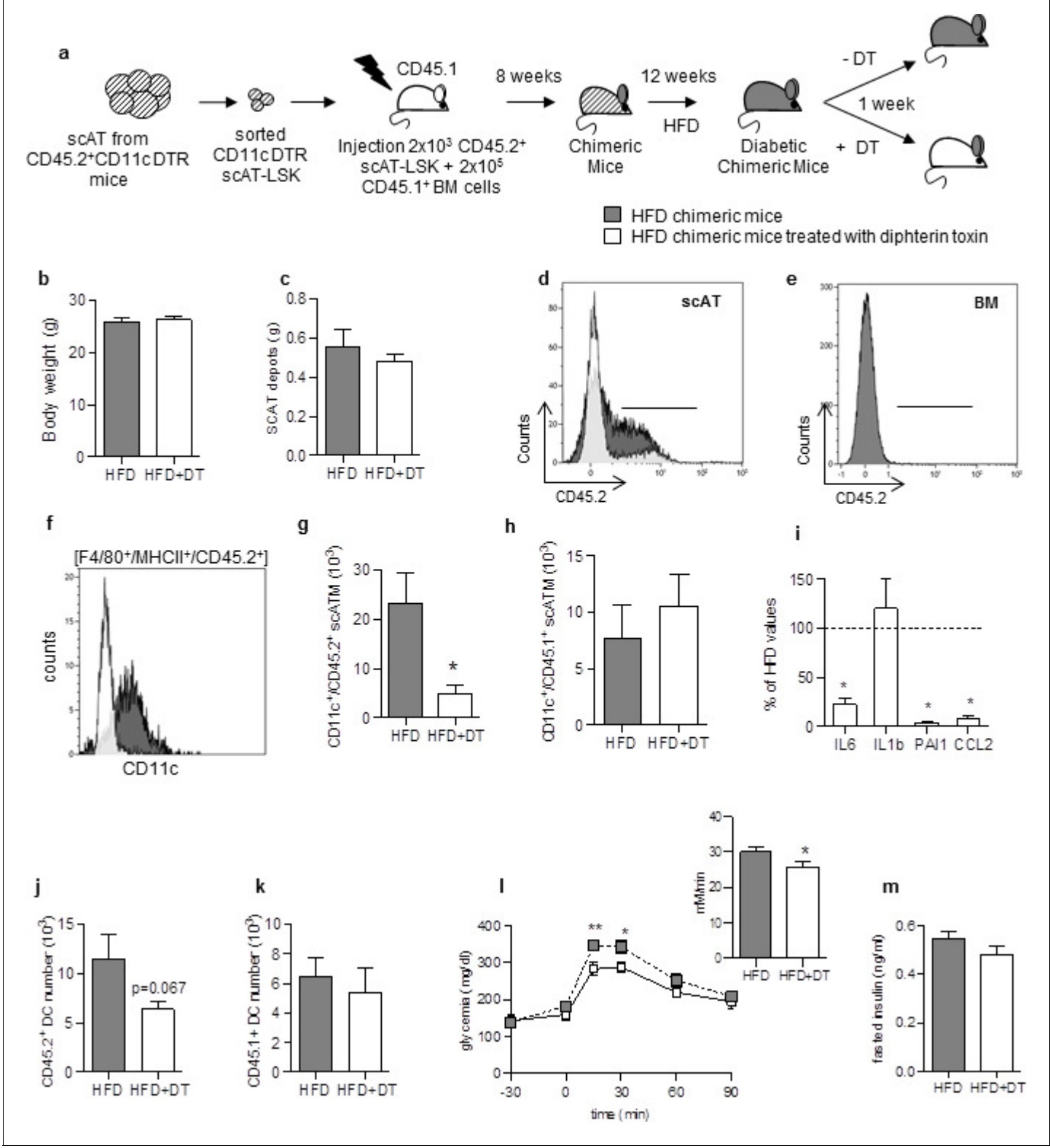

**Figure 3.** Specific depletion in inflammatory macrophages derived from scAT-LSK improves glucose metabolism in HFD mice. (a) Schematic representation of experimental procedure: lethally irradiated CD45.1⁺ recipient mice were co-injected with sorted scAT-LSK and total BM cells isolated respectively from CD45.2⁺ CD11c-DTR and CD45.1⁺ donor mice, and were then fed a high fat diet (HFD) for 12 weeks, before treatment with diphtheria toxin (DT) for 1 week. (b) Body weight and (c) scAT weight of chimeric mice measured at the end of the DT treatment. (d, e) Representative histograms of CD45.2 expression in the scAT and the BM of chimeric mice. (f–h) Representative histogram (f) and quantification of the numbers of inflammatory

*Figure 3 continued on next page*

*Figure 3 continued*

ATM derived from scAT-LSK (**g**) or BM (**h**) identified as CD45.2$^+$ or CD45.1$^+$ F4/80$^+$/MHCII$^+$/CD11c$^+$ cells respectively in HFD untreated (grey histogram) and DT-treated (white histogram) mice. (**i**) Expression of genes encoding for inflammatory cytokines analyzed by qRT-PCR in the scAT and expressed as a percent of control values obtained in untreated mice (dotted line). (**j, k**) Quantification of dendritic cells (F4/80$^-$/CD11c$^+$/MHCII$^+$) derived from scAT-LSK (**j**) or the BM (**k**). Metabolic phenotype determined with (**l**) GTT and AUC and (**m**) insulin levels. Results are expressed in mean ± sem of 4 to 8 individual animals in HFD untreated (grey symbols) or DT treated (white symbols) groups. *p<0.05; **p<0.01, nonparametric Mann-Whitney test.

The DT treatment slightly but significantly improved glucose tolerance (*Figure 3l*), without any change in fasting glucose and insulin levels (*Figure 3l and m*).

Altogether this in vivo approach demonstrates that the depletion of scAT-LSK–derived inflammatory ATM improves both inflammatory and metabolic status in HFD mice. This suggested that a modification of scAT-LSK hematopoietic activity may be one of the regulatory mechanism in this pathogenesis.

## Adoptive transfer of metabolic disease by transplantation of scAT-LSK sorted from HFD mice

To test whether scAT-LSK may play a causal role in the development of metabolic disease, repopulation assays were performed with scAT-LSK sorted from either HFD or control CD45.1$^+$ mice and transplanted in control CD45.2$^+$ recipients. Both groups of chimeric mice were then maintained on normal diet after reconstitution (*Figure 4a*).

Body and scAT weights were similar in mice transplanted with control or HFD-primed scAT-LSK (*Figure 4b and c*). Total chimerism was similar in the scAT of both groups of mice (*Figure 4d*). No CD45.1$^+$ cell was observed in the BM (*Figure 4e*).

Transplantation of scAT-LSK sorted from HFD mice induced a significant increase in scAT-LSK derived ATM in the scAT (*Figure 4f and g*), and this increased was due to an increase in CD11c$^+$ ATM number, without any change in CD206$^+$ ATM (*Figure 4g*). An increased expression of inflammatory cytokines such as IL-6, PAI-1 and CCL2 in the scAT was associated with changes in scATM content (*Figure 4h*). Mice transplanted with scAT-LSK sorted from HFD mice were glucose intolerant (*Figure 4i*), with increased fasted plasmatic glucose (*Figure 4i*) and insulin (*Figure 4j*) levels compared to mice transplanted with control scAT-LSK. A 2–3 fold decrease in glucose utilization was observed in the scAT (*Figure 4k*), demonstrating that these chimeric mice tend to be insulin resistant and corroborates the glucose tolerance data.

Altogether these data demonstrate that scAT-LSK sorted from HFD mice are able to induce the onset of metabolic disease when transplanted in vivo in control mice maintained on normal diet, via their sustained potential to generate inflammatory ATM.

## Beneficial effect of healthy AT-LSK transplantation in HFD mice

Finally, to test whether transplantation of control AT-LSK could improve the metabolic profile of HFD mice, we performed repopulation assays with scAT-LSK sorted from HFD or control donor mice grafted in HFD recipient mice. The chimeric mice were then maintained on HFD for 2 additional weeks (*Figure 5a*). No difference in body (*Figure 5b*) and scAT weights (*Figure 5c*) was observed between both groups of chimeric HFD mice. Transplantation of scAT-LSK sorted from control mice into HFD mice induced an improvement in glucose tolerance (*Figure 5d*), a decrease in fasting insulin levels (*Figure 5e*), as well as an improvement of scAT-glucose utilization rate (*Figure 5f*). These data demonstrated that transplantation of control scAT-LSK in HFD mice improves significantly scAT insulin resistance and glucose tolerance.

## Discussion

We have previously demonstrated the presence in AT of an endogenous functional hematopoietic process, based on the presence of a peculiar resident LSK population (*Poglio et al., 2012*; *Luche et al., 2015*). In steady state, AT-LSK are able to reconstitute a part of the immune cell compartment in AT or other organs except classic hematopoietic ones (*Poglio et al., 2012, 2010*). Here we show the crucial role of AT-LSK in the development of metabolic disease. Indeed, the transfer of

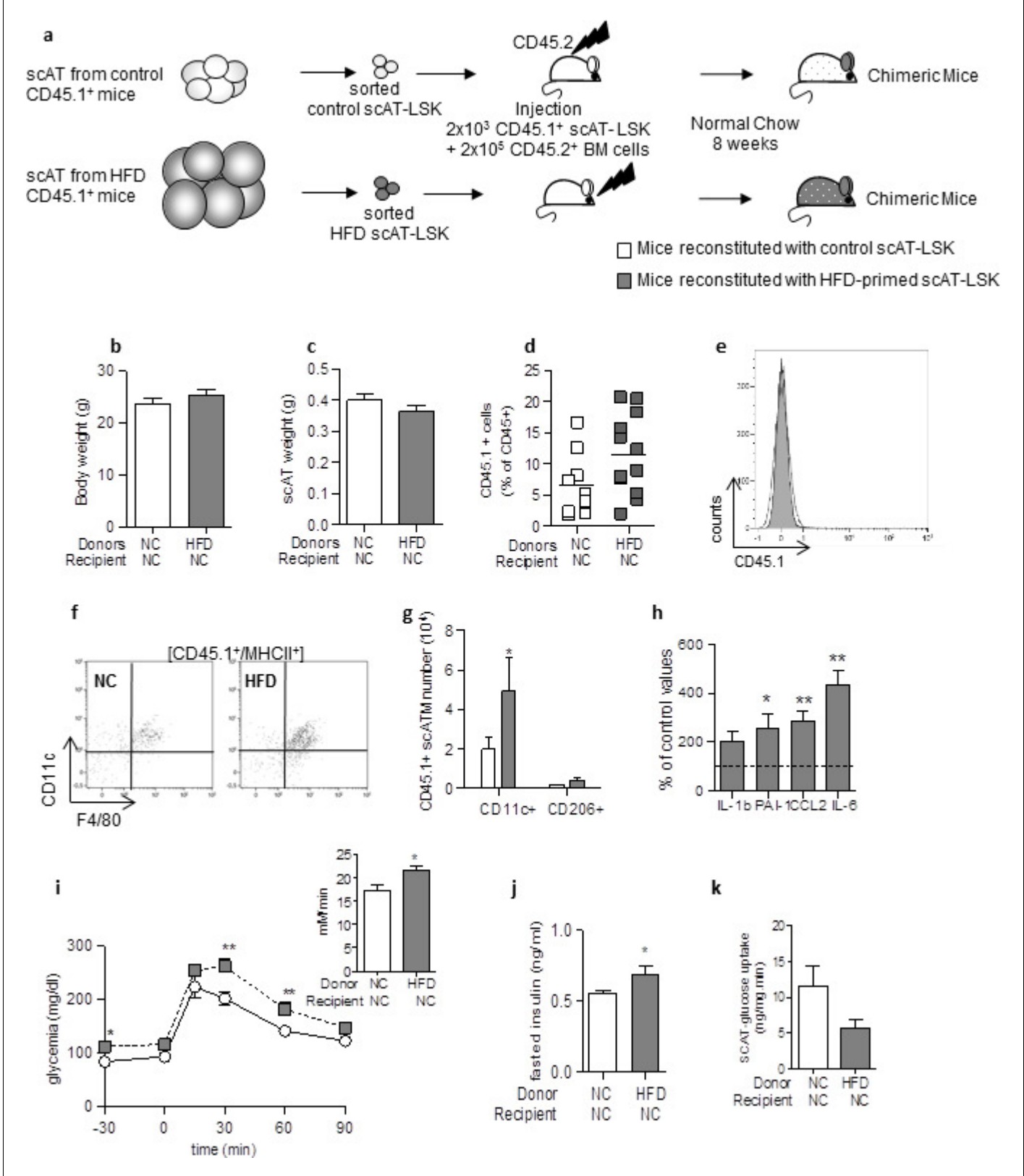

**Figure 4.** Adoptive transfer of metabolic disease with injection of scAT-LSK sorted from HFD mice into control animals. (a) Schematic representation of experimental procedure: lethally irradiated CD45.2[+] recipient mice were co-injected with sorted scAT-LSK isolated from control or HFD CD45.1[+] donor mice and total BM cells isolated from control CD45.2[+] mice. Chimeric mice were then maintained on normal chow for 8 weeks. Body (b) and scAT

*Figure 4 continued on next page*

Figure 4 continued

weights (c) measured in chimeric mice. (d) chimerism was quantified in the scAT and expressed as percent of total CD45[+] cells. (e) Representative histogram showing the absence of CD45.1[+] cells in the BM. (f) Representative dot plots showing scATM identified as CD11c[+]/F4/80[+]/MHCII[+] cells in a CD45.1[+] cell population gated on singlet live cells. (g) Quantification of CD45.1[+] pro-inflammatory (CD11c[+]) or anti-inflammatory (CD206[+]) ATM expressed in absolute numbers. (h) Expression of genes encoding for inflammatory cytokines analyzed by qRT-PCR in the scAT, and expressed as a percent of values obtained in mice reconstituted with control scAT-LSK (dotted line). Metabolic phenotype was determined with (i) GTT and AUC, (j) fasted insulin level and (k) scAT-glucose uptake. Results are expressed in mean ± sem of 5–8 individual mice transplanted with scAT-LSK sorted from either control (white symbols) or HFD (grey symbols) mice. *p<0.05, **p<0.01; nonparametric Mann-Whitney test.

scAT-LSK sorted from HFD mice into control mice induces AT inflammation and metabolic disease, demonstrating that alteration of endogenous myelopoiesis in the AT controls, at least in part, glucose homeostasis and insulin resistance.

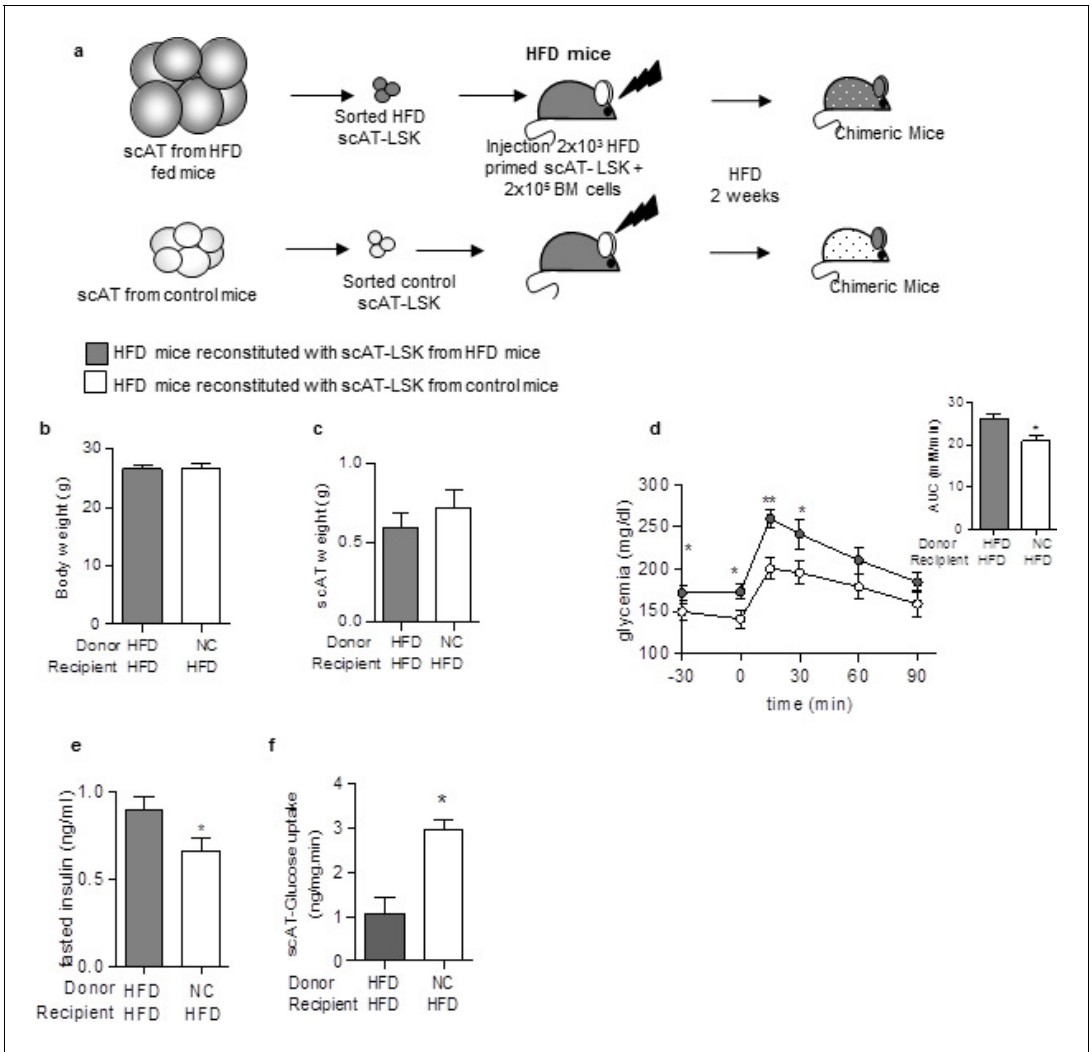

**Figure 5.** Beneficial effect of scAT-LSK transplantation in HFD mice. (a) Schematic representation of experimental procedure: HFD lethally irradiated CD45.2[+] recipients mice were co-injected with sorted scAT-LSK isolated from HFD or control CD45.1[+] donor mice and total BM cells from NC CD45.2[+] mice, and maintained on HFD for 2 additional weeks. (b) Body weight and (c) scAT weight were measured in chimeric mice at the end of the diet period. Metabolic phenotype was determined with (d) GTT and AUC, (e) fasted insulin levels and (f) scAT-glucose uptake. Results are expressed in mean ± sem of 4–14 individual animals transplanted with scAT-LSK sorted from either HFD (grey symbols) or control (white symbols) mice. *p<0.05, **p<0.01; Comparisons between groups were made with the unpaired two-sided Student's t-test (b, d–f) or the nonparametric Mann-Whitney test.

Measurement of reconstitution efficiency in all chimeric mice transplanted with scAT-LSK confirmed our previous study (*Poglio et al., 2012*). Indeed, by showing no chimerism in the BM, we confirmed that the progeny of scAT-LSK in the AT did not arise from the BM.

In diabetic context, scAT-LSK generates pro-inflammatory ATM that exhibit deleterious effects on metabolism. The induction of AT inflammation and glucose intolerance has already been described after transfer of activated antigen-presenting cells into normal mice (*Moraes-Vieira et al., 2014*). However we underline here the specific role of scAT-LSK derived-ATM, since their specific depletion in diabetic mice ameliorates metabolic features. In chimeric mice transplanted with CD11c-DTR scAT-LSK and treated with DT, the improvement of glucose tolerance was not associated with a decrease in fasting blood glucose and insulin concentrations. This set of data is in line regarding the etiology and kinetic of insulin resistance since both features are the consequences of a highly integrated set of mechanisms. Indeed, over the time course of HFD-induced hyperglycemia and insulin resistance, absorptive hyperglycemia develops first after one week then followed by fasting hyperglycemia 4 weeks later (*Garidou et al., 2015*). Fasting hyperinsulinemia follows the same kinetic. Therefore, the improvement of glucose tolerance after only one week of DT treatment was expected, while the improvement of fasting glycemia and insulinemia should require a longer period of treatment.

Our results demonstrate that the accumulation of ATM in HFD mice, which is a largely described hallmark of obesity and diabetes (*Lumeng et al., 2007*, *2008*; *Weisberg et al., 2003*; *Cani et al., 2007*; *Fujisaka et al., 2009*) originates at least in part from resident AT-LSK and occurs very early in the disease development and is mainly due to enhanced progenitor proliferation. Although our data point out an important role for these ATM derived from endogenous AT-hematopoieis, they do not rule out a possible role for other cell/macrophages subsets originating from the medullar hematopoiesis, or other cell processes. Indeed, in HFD chimeric mice, ATM derived from AT-LSK account for approximately 50% of total ATM, showing that in the AT, macrophages arise from different sources. This is in line with the recent literature showing that beside the contribution of circulating myeloid progenitors originating from the BM, in situ ATM proliferation can be responsible at least in part, for accumulating ATM in obesity (*Amano et al., 2014*). Other evidences suggest the possible involvement of preadipocytes that exhibit characteristics of macrophages (*Charrière et al., 2003*; *Cousin et al., 1999*) and that upon appropriate signals may efficiently and rapidly adapt to function as macrophages (*Boutens and Stienstra, 2016*). The specific contribution of these heterogeneous AT-immune cell subsets to development of glucose intolerance and insulin resistance will need further study.

In conclusion our data demonstrate the pivotal role of resident AT-LSK in the induction of metabolic disease, and underline the importance of ATM originating from endogenous AT hematopoietic activity. These data add thus new insights in the actual debate around macrophage ontogeny, and the contribution of different macrophage populations in tissue homoestasis (*Ginhoux and Guilliams, 2016*). Considering the abundance of AT in the body, and the generalized presence of AT-LSK in all deposits (*Luche et al., 2015*), close to most of the major vital organs, the involvement of endogenous AT hematopoietic process in disease needs further investigation.

## Materials and methods

### Animal models and dietary treatments

Experiments were performed on 6- to 8- week-old male C57BL/6J CD45.2 mice and congenic male C57BL/6J CD45.1 mice. Animals were housed in a controlled environment (12 hr light/dark cycles at 21°C) with unrestricted access to water and diet. Mice were fed with a normal chow diet (NC; 12% fat, 28% protein, and 60% carbohydrate) or a high-fat diet (HFD, 72% fat, 28% protein, and <1% carbohydrate) for 3 days to 12 weeks. For AT-CD11c$^+$ cell depletion, mice transplanted with scAT-LSK sorted from CD11c-DTR transgenic mice were injected i.p. with diphtheria toxin at a dose of 10 ng/g body weight at day −4, –2 and 0 before the GTT and flow cytometry analyses. Before removal of tissues, mice were killed by cervical dislocation.

### Isolation of AT, BM liver and muscle cells

Bone Marrow (BM) cells were flushed from the femurs with α-MEM medium (Life Technologies). Subcutaneous inguinal adipose tissue (scAT), perigonadal adipose tissue (pgAT), and liver were dissected and mechanically dissociated. scAT, pgAT and liver fragments were digested with collagenase NB4 (Serva Electrophoresis) for 30 min at 37°C under agitation. Cells from liver and AT-stroma-vascular fraction (SVF) were collected by centrifugation after elimination of undigested fragments by filtration as described previously (*Poglio et al., 2010*). Red blood cells were removed by incubation in hemolysis buffer. The quadriceps was excised and bones and tendons were removed. The muscle tissue was thoroughly minced and then digested at 37°C with 0.5 U/mL of collagenase B (Sigma) and 2.4 U/mL of Dispase II (Roche) for 30 min at 37°C under agitation. The tissue was triturated briefly by using a 18G needle. The digestion round was repeated and cell suspension was then passed through a 34 μM filter. Cells were then collected by centrifugation. Cells were counted and used for flow cytometric analysis or LSK sorting.

### Competitive repopulation assays

Competitive repopulation assays were conducted as described previously (*Poglio et al., 2012*, *2010*). Briefly, $2 \times 10^3$ Lin$^-$/Sca-1$^+$/ c-Kit$^+$ (LSK) cells sorted from the scAT or the BM of donor mice were mixed with $2 \times 10^5$ competitor BM total cells. In all the experiments, control and experimental LSK were sorted from animals of equal age. The mixed population was intra-venously injected into lethally irradiated (10Gy, 137Cs source) recipient mice of equal age. Reconstituted mice were then allowed to recover during 2 months unless otherwise indicated. To allow monitoring of the LSK progeny, we used the CD45.1/CD45.2 mismatch system. scAT- or BM-LSK expressing either CD45.1 or CD45.2 were transplanted respectively in CD45.2 or CD45.1 expressing mice together with BM cells that have the same genotype that the recipient mice. Chimerism was assessed by quantifying CD45.1$^+$ or CD45.2$^+$ cells among total CD45$^+$ cells in the SVF.

### Flow cytometry analysis and cell sorting

Flow cytometry was used to characterize adipose tissue macrophages (ATM) as described previously (*Poglio et al., 2010*). Freshly isolated SVF cells were stained in PBS containing FcR-blocking reagent. Phenotyping was performed by immunostaining with conjugated rat anti–mouse mAbs and comparing with isotype-matched control mAb (*Supplementary file 1a*). For Ki67 staining, cells were fixed and permeabilized with Cytofix/Cytoperm Fixation/Permeabilization Solution Kit (BD Bioscience) according the manufacturer's instructions. Cells were then incubated with FITC-conjugated Ki67 (Miltenyi) or isotype control mAbs for 45 min. Cells were washed in PBS and analyzed on a FACS Canto II or Fortessa flow cytometer. Data acquisition and analysis were performed using Kalusa Version 1.2 software.

For LSK cell-sorting experiments, SVF cells were stained with FITC-conjugated Ly-6A/E (Sca-1), PE-Cy7–conjugated CD117 (c-Kit) antibodies, and APC-conjugated Lineage Panel. Cells negative for lineage markers were gated, and Sca-1 and CD117 double-positive cells were sorted. The degree of the enrichment of HSC determined by flow cytometry was between 92% and 97%.

### Monitoring metabolic parameters

For the intraperitoneal glucose tolerance test (ipGTT) mice fasted for overnight or 6 hr were injected with a 20% glucose solution at a dose of 1 g glucose/kg body weight. Glycemia was determined by the use of a glucose meter on samples of blood collected from the tip of the tail vein. The aire under the curve (AUC) was determined. Plasma insulin concentration was assessed from blood collected from fasted (for 6 hr) mice by ELISA according to the manufacturer's instructions. Euglycemic hyper-insulinemic clamps were performed (*Burcelin et al., 2004*). Briefly, the mice were fasted for 6 hr and were infused at a rate of 18 or 4 mU.kg-1.min-1 of insulin for 3 hr. Simultaneously, a 20% glucose solution was infused to maintain a steady glycemia. To determine an index for the individual glucose use rate, a flash injection through the femoral vein of 30 μCi per mouse of D-2- [3 hr]deoxyglucose was performed 60 min before the end of the infusions. Individual tissue glucose uptake measurement in scAT and pgAT was determined as previously described (*Kamohara et al., 1997*).

## RNA extraction and Real-Time PCR

Total RNA from mouse tissues was isolated by Qiazol extraction and purification was done using RNeasy minicolumns. For quantitative real-time PCR analysis, RNA was reverse transcribed using a cDNA Reverse Transcription kit, SYBR Green PCR Master Mix, and 300 nmol/L primers on an Applied Biosystem StepOne instrument. Relative gene expression was calculated by the $\Delta\Delta CT$ method, normalized to 36B4, and expressed in percent of control values. Primers are listed in *Supplementary file 1b*.

## Statistical analyses

Experiments done previously were used to determine animal sample size with adequate statistical power. The number of animals used in each study is indicated in the figure legends. Studies were not randomized, and analyses were blinded to investigators. Chimeric mice exhibiting a chimerism <1% were excluded from the analysis. All results are given as means +/− sem. Normality was checked using the D'Agostino and Pearson omnibus normality test. Variance between groups was compared. Statistical differences were measured using an unpaired two-sided Student's t-test, or a nonparametric test (Mann–Whitney) when data did not pass the normality test, or when variances between groups were different. All statiscical analyses were performed in GraphPad Prism 5.0 software and a two-tailed *P* value with 95% confidence interval was acquired. $p < 0.05$ was considered as significant. The following symbols for statistical significance were used throughout the manuscript: *$p < 0.05$; **$p < 0.01$; ***$p < 0.001$.

## Study approval

Animals were maintained in accordance to guidelines of the European Community Council. All experimental procedures were done in compliance with European regulations for animal experimentation. The authors have received requested approval from their Institutional Ethic Committee, and from Ministry of National Education, Higher Education and Research (# 2691–2015110616015905) for all the experiments performed.

# Acknowledgements

The authors thank the US006/CREFRE INSERM/UPS (Toulouse, France) and specifically the Non-Invasive Exploration service for giving access to the gamma-irradiator BioBeam 8000, and the zootechnical core facility for animal care; the Toulouse RIO Imaging and flow cytometry core facility (Toulouse, France) for cell sorting, C Evra, V Marin, A Pecastaing P Guillou, A Balguerie, and A Giry for their technical assistance. Our research was supported by a grant from Aviesan/AstraZeneca, « Diabetes and the vessel wall injury » program, by a grant from the Société Francophone du Diabète, and by the French ANR Project «WAT-HEART» (ANR 16-CE14-0006-01). Rémy Burcelin is a recipient of grants from the 7th European Framework Program (Florinash), the Agence Nationale de la Recherche (ANR: Floradip, Transflora, Bactimmunodia). Céline Pomié is recipient of grant from the Société Francophone du Diabète. Virginie Robert is recipient of grant from French ANR Project «WAT-HEART».

# Additional information

## Funding

| Funder | Grant reference number | Author |
|---|---|---|
| Aviesan/AstraZeneca | | Louis Casteilla<br>Beatrice Cousin |
| Société Francophone du Diabète | | Beatrice Cousin |
| Agence Nationale de la Recherche | ANR 16-CE14-0006-01 | Beatrice Cousin |

The funders had no role in study design, data collection and interpretation, or the decision to submit the work for publication.

## Author contributions

ELu, Formal analysis, Investigation, Methodology, Writing—original draft, Project administration; VR, Formal analysis, Investigation, Writing—original draft; VC, ELa, Formal analysis, Investigation; CP, EA, Formal analysis, Investigation, Methodology; QS-A, AW, Formal analysis; AV, Formal analysis, Investigation, Writing—original draft, Experimentation; PL, Supervision, Writing—original draft, Project administration; RB, Conceptualization, Supervision, Methodology, Writing—original draft; LC, Conceptualization, Resources, Supervision, Funding acquisition, Writing—original draft, Writing—review and editing; BC, Conceptualization, Formal analysis, Funding acquisition, Investigation, Methodology, Writing—original draft, Project administration, Writing—review and editing

## Author ORCIDs

Beatrice Cousin, http://orcid.org/0000-0003-2952-4601

## Ethics

Animal experimentation: Animals were maintained in accordance to guidelines of the European Community Council. All experimental procedures were done in compliance with European regulations for animal experimentation. The authors have received requested approval from their Institutional Ethic Committee, and from Ministry of National Education, Higher Education and Research (# 2691-2015110616015905) for all the experiments performed.

# Additional files

### Supplementary files

• Supplementary file 1. Antibody (*Supplementary file 1a*) and Primer (*Supplementary file 1b*) lists.

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
