## [Decision Letter]

Thank you for submitting your article "Corrupted adipose tissue endogenous myelopoiesis initiates diet-induced metabolic disease" for consideration by *eLife*. Your article has been favorably evaluated by Sean Morrison (Senior Editor) and three reviewers, one of whom is a member of our Board of Reviewing Editors. The reviewers have opted to remain anonymous.

The reviewers have discussed the reviews with one another and the Reviewing Editor has drafted this decision to help you prepare a revised submission.

Summary:

The reviewers felt that the manuscript substantively extends your previous work in this field, providing new compelling data that AT-LSKs do function in the insulin resistance syndrome. Overall, the experiments are well done and provide a reasonable basis for the interpretations, although some further work is needed to solidify the claims.

Essential revisions:

One crucial question is the mechanism whereby donor macrophages accumulate in AT of HFD-fed mice. if macrophages originate from a resident progenitor population in the transplant model, are these cells multiplying? The authors should check whether these cells are KI67+ or better still, use direct means to determine proliferation. If this is the case, what might be the signal triggering macrophage proliferation in adipose tissue of HFD-fed mice? Definitive identification of the signal may not be possible in the context of this study but any insights you can provide along these lines would strengthen the paper.

Secondly, in order to conclude that the accumulation of ATM in the scAT is mainly due to increased CD45.1+/CD11c+ (donor), the author should also look at the changes in CD45.2+/CD11c+ (recipient) (Figure 1). Also, there is no chimerism in bone marrow following transplantation of AT-LSK, but the effect seen on metabolism could also be due to chimerism in other metabolic tissues. What is the chimerism in liver and visceral adipose tissue? (Figure 3 and Figure 4)?

Finally, CD11c is also highly expressed in dendritic cells, so using the CD11c promoter driven DTR mouse model may be problematic. At least the dendritic cell profile in WAT should be presented in the transplantation study with or without DT (Figure 2). Also in Figure 2), only quantification of F4/80+MHCII+CD11c+CD45.2+ scATM is provided. However, the F4/80+MHCII+CD11c+CD45.1+ scATM profile should also be included to allow readers to evaluate if there is any change in BM-originated cells in the WAT with or without DT treatment.

---

## [Author Response]

*Summary:*

*The reviewers felt that the manuscript substantively extends your previous work in this field, providing new compelling data that AT-LSKs do function in the insulin resistance syndrome. Overall, the experiments are well done and provide a reasonable basis for the interpretations, although some further work is needed to solidify the claims.*

We would like to thank the reviewers for their comments as this has helped us make this project even more precise and complete. New data have been included in the manuscript, as requested, and illustrated in figures. The Materials and methods section has been changed accordingly (subsection “Isolation of AT, BM liver and muscle cells”; subsection “Flow cytometry analysis and cell sorting”).

*Essential revisions:*

*One crucial question is the mechanism whereby donor macrophages accumulate in AT of HFD-fed mice. if macrophages originate from a resident progenitor population in the transplant model, are these cells multiplying? The authors should check whether these cells are KI67+ or better still, use direct means to determine proliferation. If this is the case, what might be the signal triggering macrophage proliferation in adipose tissue of HFD-fed mice? Definitive identification of the signal may not be possible in the context of this study but any insights you can provide along these lines would strengthen the paper.*

Accumulation of CD45.1+ ATM in HFD-fed chimeric mice could be due to enhanced number and/or differentiation of progenitors and/or to macrophage proliferation. As suggested by the reviewers, flow cytometry was used to quantify LSK cells, and myeloid progenitors expressing CD45.1, based on previously described phenotypes (van Galen et al., Nature, 2014), as well as Ki67+ ATM. Data are shown on Figure 2—figure supplement 1 in the revised version of the manuscript, and detailed in the first paragraph of the subsection “ATM derived from scAT-LSK accumulate in AT of HFD mice due to early proliferation of progenitor cells”. Briefly, after 3 months of HFD, the number of total CD45.1+ LSK and myeloid progenitor cells were similar to NC values. The rate of proliferating ATM is similar in both groups of mice. This result suggests that the dynamic phase of the increase in CD45.1+ATM number observed after 3 months of HFD occurs earlier in the pathogenesis. Indeed, after a few days of HFD, CD45.1+ LSK and myeloid progenitor cell numbers were increased due to an enhanced proliferation, whereas the number and the proliferation of CD45.1 ATM were not yet changed. These results have been included in Figure 2 and in the aforementioned paragraph, and in the fourth paragraph of the Discussion. We were however unable to identify the signal triggering the proliferation of LSK cells and myeloid progenitors.

*Secondly, in order to conclude that the accumulation of ATM in the scAT is mainly due to increased CD45.1+/CD11c+ (donor), the author should also look at the changes in CD45.2+/CD11c+ (recipient) (Figure 1).*

To conclude that the accumulation of ATM in the scAT is mainly due to increased donor cells (CD45.1+), we have quantified as requested by the reviewers, the number of CD45.2+ ATM. This has been added on Figure 1, and in the fourth paragraph of the subsection “Inflammatory ATM derived from subcutaneous (sc) AT-LSK differentiation accumulate in HFD mice”. The graph shows that the number of CD45.2+ cells is similar in the scAT of HFD and NC chimeric mice.

*Also, there is no chimerism in bone marrow following transplantation of AT-LSK, but the effect seen on metabolism could also be due to chimerism in other metabolic tissues. What is the chimerism in liver and visceral adipose tissue? (Figure 3 and Figure 4)?*

As mentioned by the reviewers, the HFD-induced metabolic effect in chimeric mice may also be due to chimerism in other metabolic tissues. We therefore determined CD45.1 chimerism in perigonadal AT (pgAT), liver and muscle. As shown on Figure 1, chimerism in liver and muscles is very low, suggesting no significant impact of CD45.1+ cells in these organs. In contrast, chimerism in pgAT reaches 15% in NC chimeric mice and is slightly decreased in HFD group (Figure 1). We thus analyzed CD45.1+ pgATM number and the expression of pro-inflammatory cytokines in this fat pad. Both CD11c+ and CD206+ ATM numbers originating from donor mice were similar in NC and HFD mice (Figure 1—figure supplement 2). Accordingly, the expression of pro-inflammatory cytokines in the pgAT was not significantly modified by the HFD (Figure 1—figure supplement 2). These results have been included in the third and fourth paragraphs of the subsection “Inflammatory ATM derived from subcutaneous (sc) AT-LSK differentiation accumulate in HFD mice”, and demonstrate that perturbation of endogenous myelopoiesis in the scAT may be the main cause of the HFD-induced metabolic change.

*Finally, CD11c is also highly expressed in dendritic cells, so using the CD11c promoter driven DTR mouse model may be problematic. At least the dendritic cell profile in WAT should be presented in the transplantation study with or without DT (Figure 2). Also in Figure 2), only quantification of F4/80+MHCII+CD11c+CD45.2+ scATM is provided.*

As mentioned by the reviewers, CD11c is also highly expressed in dendritic cells (DC). The number of donor- and recipient-DC was therefore evaluated in the scAT of chimeric mice, based on the phenotype F4/80-/CD11c+/MHCII+, and included in Figure 3. The CD45.2+ DC number (donor origin) was slightly decreased in treated mice, whereas the CD45.1+DC number remained unchanged. However, since the HFD did not increase the CD45.1+ DC number in the scAT of chimeric mice, we think that this decrease may not play a key role in the pathogenesis. These results have been included in the second paragraph of the Results subsection “Specific depletion of ATM derived from scAT-LSK improves glucose metabolism”.

*However, the F4/80+MHCII+CD11c+CD45.1+ scATM profile should also be included to allow readers to evaluate if there is any change in BM-originated cells in the WAT with or without DT treatment.*

To evaluate whether the DT treatment induces changes in the BM originating ATM in the scAT, the number of CD45.1+/CD11c+ ATM has been included in Figure 3 and in the second paragraph of the subsection “Specific depletion of ATM derived from scAT-LSK improves glucose metabolism”. As expected, DT treatment induces no modification of BM-originated ATM and DC.